# Assessment of Shallow Groundwater Contamination Resulting from a Municipal Solid Waste Landfill—A Case Study in Lianyungang, China

**Ge Chen** [1,*] **, Yajun Sun** [1]**, Zhimin Xu** [1]**, Xuekai Shan** [2] **and Zhengliang Chen** [2]

[1] School of Resources and Geosciences, China University of Mining and Technology, Xuzhou 221116, China; syj@cumt.edu.cn (Y.S.); xuzhimin@cumt.edu.cn (Z.X.)
[2] Jiangsu Keyida Environmental Protection Technologies Co. Ltd., Yancheng 224005, China; sxk@163.com (X.S.); czliang068@163.com (Z.C.)
\* Correspondence: cg5@cumt.edu.cn

**Abstract:** Groundwater contaminations based on the release and transportation of leachate from municipal solid waste (MSW) landfills are a potential hazard to the ecosystem and its inhabitants. In this study, nine chemical compositions of groundwater quality were collected and analysed from 16 monitoring wells and two ponds around the Diaoyushan MSW landfill in the north of Jiangsu Province, China. Multiple analyses were performed to assess the redox conditions and the groundwater environment. It was indicated that the landfill was in a low and stable biodegradability phase, and the most influential phase was the initial stage of the landfill site; the leachate leakage was the principal pollution source (49.18%) for the local groundwater environment. Artificial drainage of Dongdasha village expanded the contaminant plume scopes and deteriorated water quality further. The polluted groundwater area was provided with high concentrations of total hardness, $Cl^-$, $SO_4^{2-}$, total dissolved solids (TDS) and Pb.

**Keywords:** factor analysis; leachate; shallow groundwater contamination; municipal solid waste landfill

## 1. Introduction

In a large proportion of developing countries, municipal solid waste (MSW) is disposed simply and unscientifically in various kinds of geological areas, which results in serious environmental and ecological risk to the surrounding water, air and soil [1]. According to the World Bank report, it estimates that the generation rate of MSW is expected to increase to 0.7 kg/capital/day and 376,639 tonnes/day by 2025. There are also considerable numbers of municipal solid waste landfills around cities in China [2].

Due to accelerating economic development, a number of contaminant leakages from landfills pose an especially large threat to groundwater quality, human health and ecosystems [3,4]. Based on the Chinese Statistical Yearbook, there were 654 landfills with a capacity of 120.38 million tons in 2017. However, the leakage of leachate from landfills has already caused groundwater and soil pollution according to previous scholarly research from field investigations and environmental assessment reports in China [5]. The generalisability of much published research on the issue of MSW pollution has shown that the significant groundwater contamination leaking from landfills includes conventional ions, such as chloride ($Cl^-$), ammonium ($NH_4^-$), total hardness, total dissolved solids (TDS), nitrate ($NO_3^-$), nitrite ($NO_2^-$) and sulfate ($SO_4^{2-}$), as well as trace metals [6–11]. Different from industrial chemical pollution [12], the leachate generally contains high concentrations of conventional ions, dissolved organic components, inorganic substances and trace metals, which are the major environmental contaminants due to their geoaccumulation and high toxicity. The chemical parameters of leachate

from landfills vary significantly due to the climatic conditions, domestic waste characteristics, collection method, disposal methods, landfill technology and so on [13–16]. Nevertheless, to some extent, leachate contamination of local groundwater is the most serious environmental impact of MSW landfills [17], especially those with no bottom liner and the collection system will aggravate the potential deterioration of the groundwater environment [18].

Over the past few years, landfill water pollution index, leachate pollution index, Nemerow index, water quality index, chemometric expertise, inverse modelling, statistical analysis and spatial variation [19,20] have been used to evaluate the physicochemical characteristics of the contaminated groundwater, and to simulate the possible environmental impacts. However, most of the studies lack the integrative and comprehensive analysis to investigate the pollution sources and transport paths of the shallow groundwater pollution, which is a result of the leachate leakage from landfills. Thus, the present study presents fundamental research of nine measured chemical components from 16 groundwater monitoring wells and two surface water ponds. Factor analysis and the Nemerow index of data will be integrated into the geographic information system (ArcGis 10.4 (Esri, Los Angeles, CA, USA)) to conduct a thorough investigation of the hydrogeochemical characteristics and spatial distribution of the polluted groundwater area. The contribution of the landfill to groundwater quality of the study area and the mechanism of how to influence the groundwater environment from the initial stage to the present and future will be studied and discussed. This will be helpful in proposing an appropriate scientific approach for closure of MSW landfill sites and remediation in the study area.

## 2. Site Description and Hydrogeological Condition

### 2.1. Site History and Condition

Diaoyushan municipal solid waste (MSW) landfill has an area of 66,714 m$^2$ and is situated in the northeast of Jiangsu Province, as seen in Figure 1, and started operation in August 1995 with a total capacity of 2.38 million cubic meters. Its designed service lifetime was extended at the end of 2016 to include an additional 2.497 and 0.096 million cubic meters of MSW and fly ash, respectively. Fly ash was sourced from the incineration of MSW in Chenxing Environment Limited Company and reached the standard of Chinese "Standard for Pollution Control on the Landfill Site of Municipal Solid Waste" (GB 16889-2008). After the treatment of chelation and solidification, its chemical composition concentration of lixivium, such as Pb 0.019mg/L and As 0.011 mg/L, were all below the standard (Pb 0.25 mg/L and As 0.30 mg/L). Not only domestic solid waste, but also their incineration fly ash, was filled in the west of this landfill. Considering various historical factors, this landfill excluded an impermeable artificial system as well as leachate drainage facilities [21]. Three waste dams, the leachate collection pond and the pretreatment station at the eastern side of the landfill were not built until the end of 2007, as seen in Figure 2, and the pretreatment station, with a capacity for 70 m$^3$/day, was abandoned in January 2015. After that, the leachate would flow automatically from the waste dam into the collection pond and would be periodically transported to the sewage disposal plant of the Chenxing Environment Limited Company by truck. Owing to the influence of poor management and funding, the surrounding and downstream groundwater environments of this landfill had already been contaminated from the leakage of leachate according to the groundwater quality tests from 1995 to 2017 for the adjacent groundwater monitoring wells around Dongdasha Village.

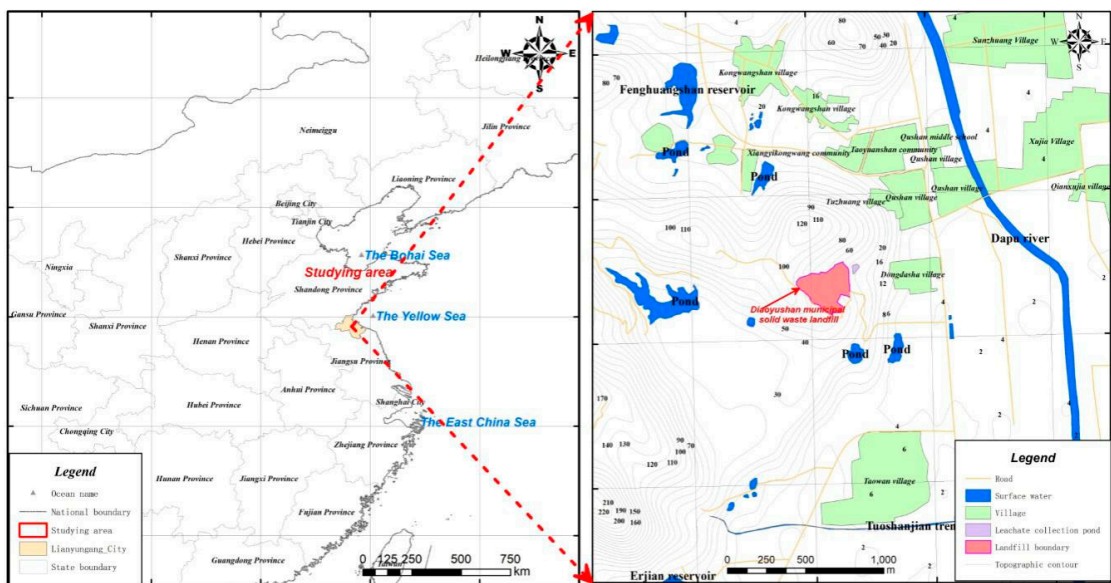

**Figure 1.** The location of the study area in northern China and Diaoyushan municipal solid waste (MSW) landfill located in the cove of Jinping Mountain.

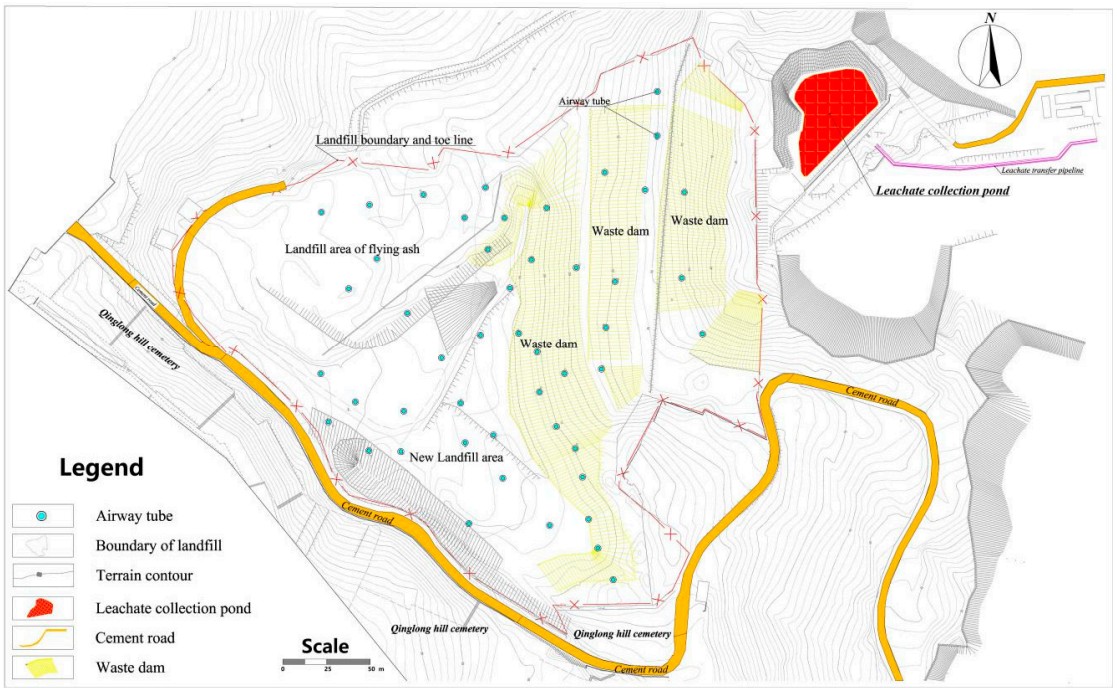

**Figure 2.** Layout for the Diaoyushan MSW landfill including the waste dams, leachate collection pond, airway tube and various geographic elements.

The clean water pipeline network and sewage pipelines for each household were supplied and no drinking wells were used in Dongdasha Village since the groundwater pollution was firstly detected by the surrounding monitoring wells. Only several irrigation wells and domestic wells were reserved as the monitoring wells.

### 2.2. Geological Background and Hydrogeological Condition

Diaoyushan MSW landfill is located in the cove between Jinping Mountain and Huaiheding. The whole topography presents the shape of a "dustpan", so the eastern open area becomes the only flow direction of the water. The Jinping Mountain reverse anticline with one NNE axial direction

extends from Jinping Mountain past Dapu Town to the estuary of the Linhong River. The strata of this study area include the Qushan Formation, Jinping Formation and Quaternary sedimentary strata [22,23], as seen in Figure 3, of which the Qushan Formation forms the core of the anticline and the Jinping Formation is distributed along with the anticline limbs. NW and NWW directional faults offset this discontinuous anticline such that the southeast limb is normal fault and the northwest is a reversed fault. The substratum of the garbage body for Diaoyushan MSW landfill is Jinping Formation, which is distributed around the east, south and west of Jinping Mountain with thicknesses exceeding the 1780 m. The main lithologies include micacite, fine-grained phosphorite, biotite amphibole schist, dolomitic plagioclase gneiss, dolomitic quartzite and lenticular marble [24]. The natural liner of the landfill is the granite from Jinping Formation due to its very poor permeation. The design of the landfill in 1995 was coupled with the geological condition. Two ditches distanced 20 m and 40 m from the toe line were constructed. The lower layers of the waste dams were all grouted to prevent the leakage of leachate.

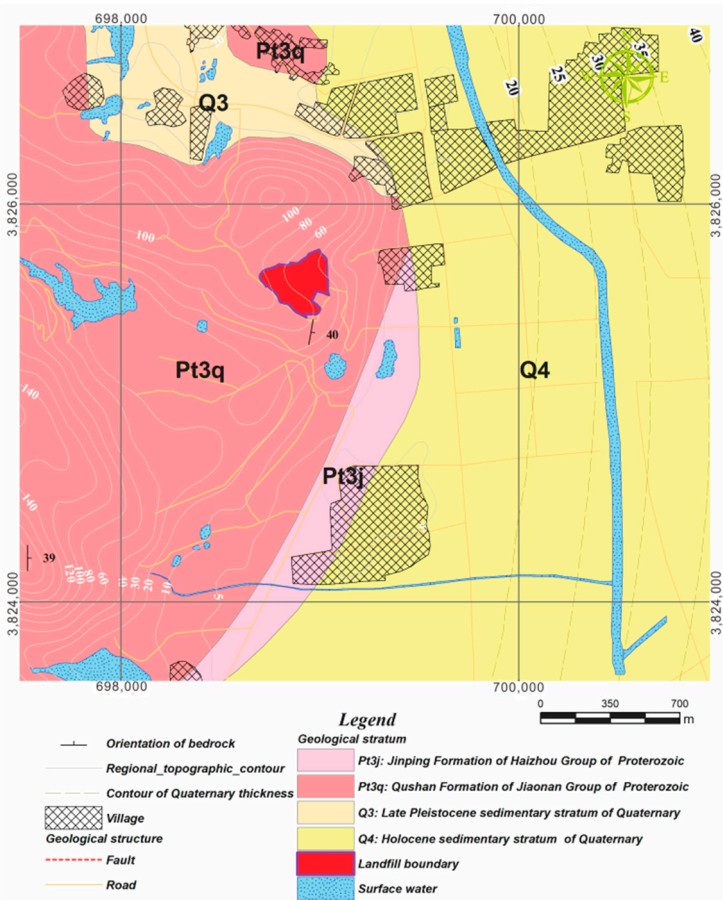

**Figure 3.** Geological map of the study area.

The main lithologies of Qushan Formation include blended granite-gneiss and plagioclase gneiss, which are distributed in the south of the landfill and underlie the Quaternary sedimentary strata. Strata have a SE inclination, dipping from 30° to 50°, and an oblique unconformable contact with the Quaternary sedimentary strata. The Qushan Formation outcrops around Jinping Mountain where it experiences strong migmatisation.

Hydrogeologically, this study area falls within the piedmont alluvial plain of Jinping Mountain. The strata of MSW landfill include fly ash, garbage soil, several covering soil layers and the moderately weathered granite (Qushan Formation). According to the water pressure tests for the granite and lab analysis for soil samples of the landfill in 2016, the garbage soil layers with the hydraulic conductivity ranges from $1.72 \times 10^{-4}$ cm/s to $1.47 \times 10^{-3}$ cm/s, which are dominated by pore groundwater. The maximum depth of the waste body is 34 m. With increasing depth, the granite is transformed from

completely or moderately weathering to weak weathering or fresh granite. Moderately weathered granite with grain sizes ranging from 0.2 mm to 0.5 mm are well developed with joint structures within the shallow depth of 15 m, and the hydraulic conductivities are varied from $9.98 \times 10^{-6}$ cm/s to $1 \times 10^{-4}$ cm/s. The dense joints zones are usually distributed at intervals of approximately 20 m, and with increasing depth, there is less joint development. The hydraulic conductivity is from $1 \times 10^{-7}$ cm/s to $3.03 \times 10^{-6}$ cm/s when the depth is 30 m, and fewer joints existed, so this unweathered section is defined as aquitard or the approximate barrier. In this landfill site, a joint oriented at 290° and tilting at 20° is consistent with the opening direction of the cove and cuts off the other joints, so the leachate will flow into the collecting pond through this joint based on no evidence of leakage points found in the bottom of the landfill and with rainfall as the only water supply source.

The Quaternary sediment is situated around the alluvial-proluvial area of Jinping Mountain, with thickness ranging from 0 to 25 m in the study area. According to the log of monitoring wells GW05 and GW06, the Quaternary sediment is composed of yellow clay with several thin grey silty clay interlayers and one layer of intermediate yellow fine sand (the thickness is 2.50 meters) within a depth of 15 m. The thickness of Quaternary sediment at the south boundary of Dongdasha Village is over 15 m. The bedrock fissure water flows through the flowing fracture or fissure zone to replenish the unconsolidated pore water in the Quaternary sediment, and on the other hand, the porous medium of the Quaternary sediment will purify, filter and transfer the groundwater. Furthermore, the overall flowing direction of groundwater is in line with the slope of the terrain, but owing to the blockade of Dongdasha village, parts of the groundwater flow to the east and southeast.

## 3. Materials and Methods

### 3.1. Sampling Sites and Chemical Test

In order to assess the depth and area of pollution resulted from MSW landfill, five wells (GW05 to GW09) were constructed from 25th June to 3rd July in 2017 and water pressure tests were conducted. The leachate sample was collected from the collection pond and tested to assess its characteristics and stability. Those groundwater samples from the surrounding monitoring wells, which were distributed around the downstream of the Diaoyushan MSW landfill site in order to monitor the contamination plume of the aquifer, were successively collected and tested, including five groundwater samples (1#, 2#, 3#, 4# and 5#) in the March of 2014, five pore phreatic groundwater samples collected on the 14th February of 2017, six pore phreatic groundwater samples (1#, 2#, 2-1#, 4#, GW05 and GW06 ) and four bedrock fissure groundwater samples (3#, GW07, GW08 and GW09 ) collected on 6 July 2017. The two surface ponds water samples (P-1# and P-2#) were also collected on 23 July 2017, as seen in Figure 4 and Table 1. Due to the discontinuous monitoring and restrained information, parts of groundwater quality data were missed in 2015 and 2016.

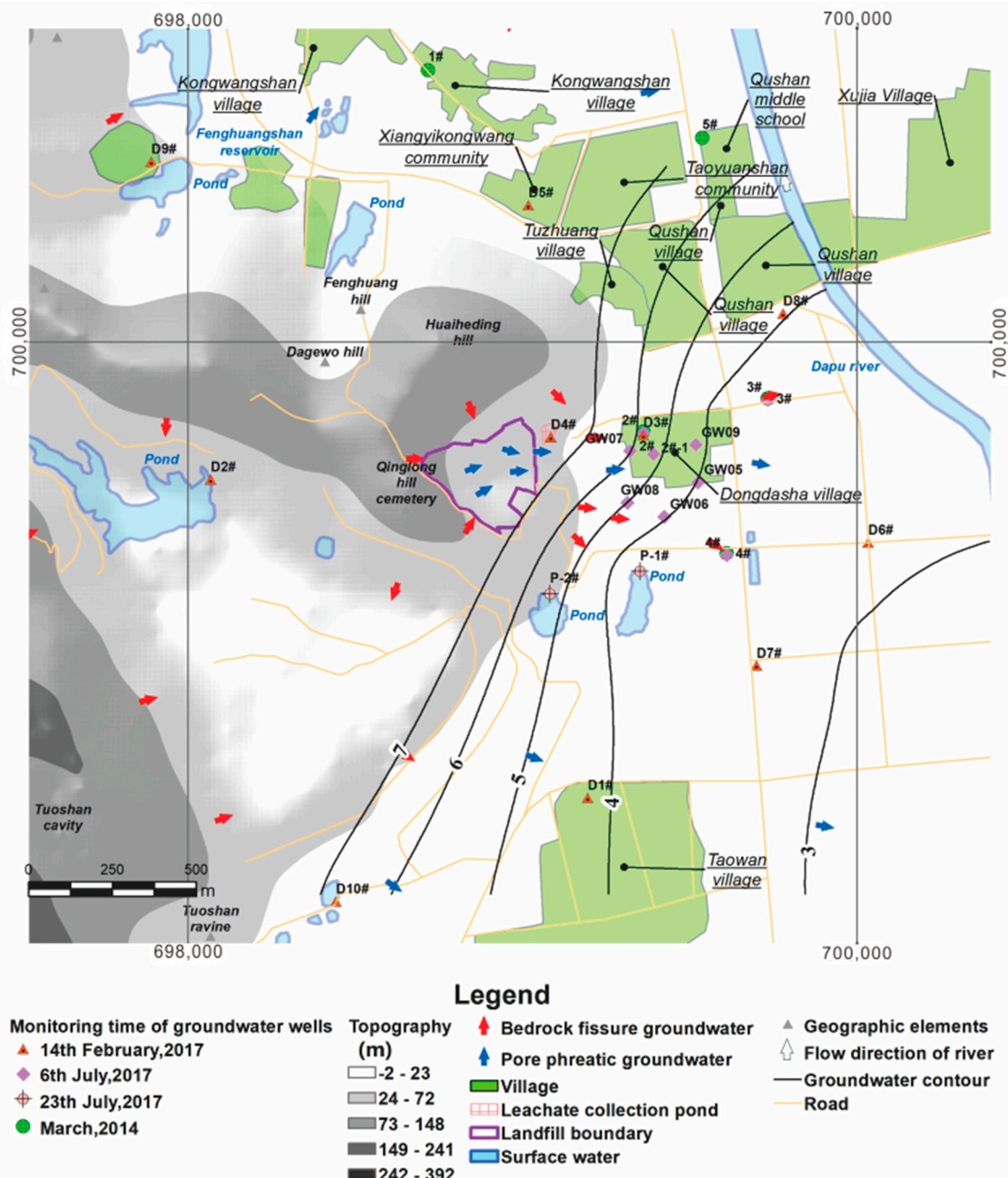

**Figure 4.** Locations of groundwater monitoring wells and the surrounding geographical environments in the study area.

All the samples, including leachate, groundwater and surface water, were collected, preserved and analysed based on the Chinese specification HJ/T166-2004 of the "Technical specification for environmental monitoring of groundwater", HJ/T168-2010 of the "Environmental monitoring Technical guideline on drawing and revising analytical method standards" and GB/T14848-2017 of the "Standard for groundwater quality" in China. GB/T14848-2017 is the available standard supported for the national groundwater pollution survey and evaluation. To ensure the integrity of analysis data, each groundwater sample was taken below a water level of 1.0 m after sufficient pumping commenced to ensure the stability of water quality, which was conducted by the measured parameters at the field, including colour, pH, turbidity and water temperature. Parallel samples were prepared to improve data quality. According to the continuity of sampling test analysis and the representativeness of test results, nine important chemical components' data, including $COD_{Mn}$ (potassium permanganate is

used as the oxidising agent), $NH_3$-N, $Cl^-$, $SO_4^{2-}$, $NO_2^-$, TDS, total hardness, Mn and Pb, were chosen to contribute to the data analysis, as seen in Table 1. They showed the overall condition of water quality in the study area, including the physical and chemical properties, organic properties and trace metal conditions as well as all the excessive chemical compositions in the water. The measurement of $BOD_5$ was tested by the dilution and inoculation method from HJ505-2009. $COD_{Mn}$ was observed under the specification GB/T 11892-1989. All the data were evaluated by semivariogram and mathematical statistics before analysis and discussion. We found that the linear model provided the best fit and the system error of the general Kriging interpolation was minimal, so the general Kriging method was used to describe the spatial characteristics of ions and compositions.

### 3.2. Factor Analysis and the Nemerow Index Calculation

Factor analysis is a multivariate analysis technique widely used to seek the relationship and difference between measured variables. It is generally defined as Equation (1) [25]

$$Z = AF + U <=> \begin{pmatrix} z_1 \\ z_2 \\ \vdots \\ z_p \end{pmatrix} = \begin{pmatrix} a_{11} & a_{12} & \cdots & a_{1m} \\ a_{21} & a_{22} & \cdots & a_{2m} \\ \cdots & \cdots & \cdots & \cdots \\ a_{p1} & a_{p2} & \cdots & a_{pm} \end{pmatrix} \begin{pmatrix} f_1 \\ f_2 \\ \cdots \\ f_m \end{pmatrix} + \begin{pmatrix} u_1 \\ u_2 \\ \cdots \\ u_m \end{pmatrix}, m \le p \tag{1}$$

where the measured data with $p$ variables are $Z = (z_1, z_2, \cdots\cdots, z_p)$, $A = (a_{11}, a_{12}, a_{13}, \cdots\cdots, a_{pm})$ is the factor loading matrix, $F = (f_1, f_2, f_3, \cdots\cdots, f_p)$ is the common factor, $U = (u_1, u_2, \cdots\cdots, u_p)$ is the sole variable for $p$ and $m$ is the number of all common factors.

The Nemerow index method is employed to assess the groundwater quality of this study area. The equation of the Nemerow index ($P$) is set as Equation (2) [26]:

$$P = \sqrt{\frac{P_{\max}^2 + P_{avg}^2}{2}} \tag{2}$$

where $P$ is the Nemerow index for the $i$th contaminant; $P_{\max}$ is the maximum value of $P_i$ for all monitoring samples, and $P_{avg}$ is the average value of $P_i$ for all monitoring samples as calculated by Equation (3):

$$P_{avg} = \frac{1}{n}\sum_{i=1}^{n} P_i \tag{3}$$

where $P_i$ is the contaminant index for the $i$th component calculated according to Equation (4):

$$P_i = \frac{C_i}{S_i} \tag{4}$$

where $C_i$ is the monitoring value for the $i$th component in the groundwater sample, and $S_i$ is the maximum value for the $i$th component in the groundwater sample specified by GB/T14848-2017 "Standard for groundwater quality" with a confirmed standard classification from I to V, corresponding to the score from 0, 1, 3, and 6 to 10.

### 3.3. Statistical and Spatial Distribution Analysis

In this study, data of nine important chemical components in the samples were tabulated and analysed using the Statistical Package for Social Sciences (SPSS) 17.0 software (IBM, Chicago, IL, USA). Factor analysis (FA) was carried out to determine the similarities and differences present in the samples to assess and classify the influence of the pollution sources in the study area.

Spatial distribution characteristic analysis of each contaminant was a dominant approach to assess the pollution degree and scope [27]. The chemical data, including $NH_3$-N, $COD_{Mn}$, Pb, total hardness,

$Cl^-$, $SO_4^{2-}$, TDS and Mn, were incorporated into the platform of the ArcGis 10.4 environment using the general Kriging interpolation. The regional variable, semivariance and variogram are taken in consideration by the general Kriging interpolation algorithm, which was the optimal and unbiased estimation of interpolation after the comparison with the methods of Nearest Neighbor [20] and Inverse Distance to a Power.

## 4. Results and Discussion

### 4.1. Analysis of Groundwater Quality in the Study Area

The sample of leachate was obtained in July of 2017 for comparison of groundwater quality and for quantitative analysis. The value of pH, $COD_{Mn}$, $BOD_5$, total nitrogen and $NH_3$-N were, respectively, 8.73, 754mg/L, 238mg/L, 829mg/L and 658mg/L.

Most of the polluted groundwater was classified as Cl–Ca or $SO_4$–Na type, as shown by the Piper diagrams in Figure 5a; the polluted groundwater possessed high concentrations of $Cl^-$ and $Na^+$ + $K^+$, but relatively low concentrations of $Ca^{2+}$ and $SO_4^{2-}$. These results showed that the polluted groundwater was in a weak acid environment with high $Cl^-$ and TDS, as seen in Figure 5b.

**Table 1.** Chemical composition analysis of groundwater samples from the investigated monitoring wells and surface water samples from the surrounding ponds (Unit: mg/L).

| Well ID (Survey Time) | Location | Water Type of Well/Pond | Distance from Leachate Collecting Pond (m) | Hydro-Chemistry Type | pH Value | NH$_3$-N (mg/L) | COD$_{Mn}$ (mg/L) | NO$_2^-$ (mg/L) | Total Hardness | Cl$^-$ (mg/L) | SO$_4^{2-}$ (mg/L) | Pb (mg/L) | TDS (mg/L) | Mn (mg/L) | Nemerow Index |
|---|---|---|---|---|---|---|---|---|---|---|---|---|---|---|---|
| 1# (March 2014) | Shantoujiulou | Pore phreatic groundwater | 1141 | SO$_4$–Na | 7.02 | 0.10 | 3.8 | 0.004 | 376 | 100 | 144 | 0.00573 | 715 | 0.0600 | 4.54 |
| 2# (March 2014) | Dongdasha village | Pore phreatic groundwater | 283 | SO$_4$–Ca | 6.87 | 0.12 | 4.0 | 0.008 | 468 | 77 | 150 | 0.00454 | 773 | 0.0302 | 4.56 |
| 2# (July 2017) | | | | HCO$_3$–Ca | 7.43 | 0.20 | 2.0 | 0.035 | 240 | 83 | 68 | 0.00125 | 385 | 0.0092 | 2.26 |
| 2#-1 (July 2017) | Dongdasha village | Pore phreatic groundwater | 320 | Cl–Ca | 7.61 | 0.29 | 2.2 | 0.041 | 505 | 258 | 107 | 0.00125 | 632 | 0.0605 | 4.66 |
| 3# (March 2014) | Unicom tower | Deep bedrock fissure groundwater | 663 | Cl–Na | 7.17 | 0.72 | 3.9 | 0.013 | 3490 | 7960 | 356 | 0.00366 | 16,400 | 3.8300 | 8.18 |
| 3# (July 2017) | | | | Cl–Na | 7.47 | 2.36 | 3.0 | 0.004 | 4860 | 5710 | 353 | 0.10400 | 8440 | 0.0089 | 8.49 |
| 4# (March 2014) | Office | Pore phreatic groundwater | 644 | Cl–Na | 6.96 | 0.42 | 5.9 | 0.009 | 3200 | 6220 | 553 | 0.00039 | 13,600 | 4.6600 | 8.05 |
| 4# (July 2017) | | | | Cl–Na | 7.52 | 0.59 | 2.1 | 0.016 | 4090 | 5750 | 511 | 0.08910 | 8360 | 0.0072 | 8.15 |
| 5# (March 2014) | Qushan middle school | Pore phreatic groundwater | 992 | Cl–Na | 7.06 | 0.40 | 1.7 | 0.006 | 591 | 1100 | 162 | 0.00248 | 2190 | 0.9550 | 7.52 |
| GW05 (July 2017) | Farmland | Pore phreatic groundwater | 470 | Cl–Na | 6.87 | 1.24 | 6.0 | 1.400 | 392 | 394 | 153 | 0.00125 | 1220 | 1.0400 | 7.69 |
| GW06 (July 2017) | Farmland | Pore phreatic groundwater | 428 | Cl–Na | 6.69 | 1.07 | 3.6 | 0.009 | 492 | 292 | 255 | 0.00125 | 986 | 0.0200 | 4.77 |
| GW07 (July 2017) | Farmland | Deep bedrock fissure groundwater | 248 | SO$_4$–Na | 7.44 | 0.94 | 9.9 | 0.474 | 248 | 256 | 356 | 0.00420 | 492 | 0.0038 | 7.59 |
| GW08 (July 2017) | Farmland | Deep bedrock fissure groundwater | 315 | Cl–Ca | 7.52 | 1.05 | 7.6 | 1.350 | 742 | 420 | 297 | 0.00125 | 1130 | 0.8240 | 8.21 |
| GW09 (July 2017) | Orchard | Deep bedrock fissure water | 441 | HCO$_3$–Ca | 7.54 | 1.59 | 2.4 | 0.008 | 224 | 95 | 78 | 0.00125 | 401 | 0.0028 | 7.17 |
| D1# (Feb. 2017) | Taowan village | Pore phreatic groundwater | 1100 | HCO$_3$–Na | 8.34 | 0.05 | 0.6 | 0.008 | 222 | 122 | 132 | 0.01130 | 1004 | 0.0860 | 4.47 |
| D2# (Feb. 2017) | Diaoyu hill | Pore phreatic groundwater | 1020 | Cl–Mg | 7.46 | 0.41 | 1.4 | 0.008 | 320 | 118 | 67 | 0.00125 | 450 | 0.0020 | 2.31 |
| D4# (Feb. 2017) | Near leachate collection pond area | Shallow bed rock fissure groundwater | 16 | Cl–Na | 8.27 | 6.97 | 5.5 | 0.010 | 728 | 333 | 127 | 0.00125 | 1477 | 0.0860 | 7.73 |
| D5# (Feb. 2017) | Taoyuanshan community | Pore phreatic groundwater | 678 | HCO$_3$–Na | 8.24 | 7.92 | 3.8 | 0.009 | 406 | 110 | 134 | 0.00125 | 763 | 0.0840 | 7.32 |
| P-1# (July 2017) | 1# pond | Surface water | 500 | SO$_4$–Ca | 6.52 | 1.03 | 6.4 | 0.021 | 571 | 230 | 478 | 0.00125 | 812 | 0.2000 | 7.82 |
| P-2# (July 2017) | 2# pond | Surface water | 485 | Cl–Na | 6.54 | 0.34 | 5.8 | 0.006 | 253 | 314 | 109 | 0.00125 | 694 | 0.0350 | 4.56 |
| The III standard concentration in GB/T14848-2017 (China) | | | / | / | 6.5–8.5 | 0.50 | 3.0 | 1.000 | 450 | 250 | 250 | 0.01000 | 1000 | 0.1000 | / |

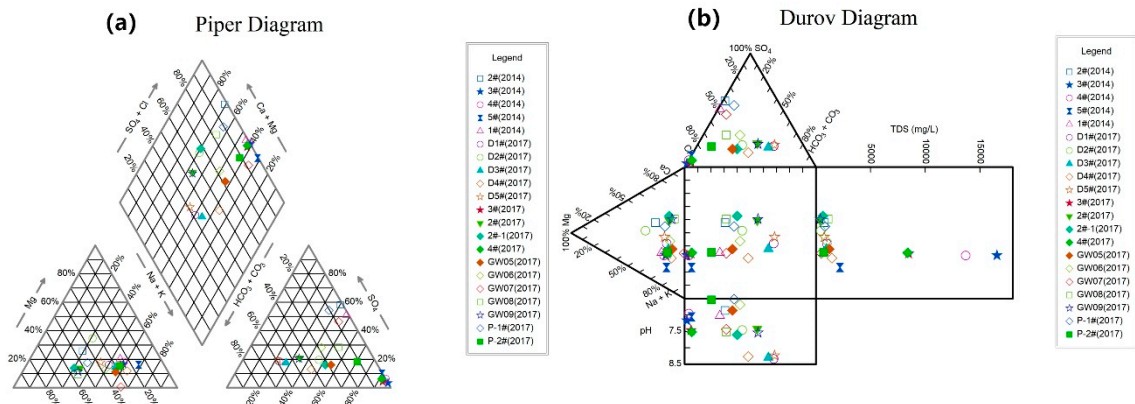

**Figure 5.** Piper and Durov diagrams for water quality of all monitoring wells. (**a**) Piper Diagram, (**b**) Durov Diagram.

$BOD_5/COD_{Mn}$ is commonly employed as an indicator of the age of MSW landfill sites [28,29]. According to the environmental assessment report of 1995, the ratio of $BOD_5/COD_{Mn}$ for leachate (512 mg/L and 1938 mg/L, respectively) and the groundwater of the 2# monitoring well (0.9 mg/L and 3.4 mg/L, respectively) was 0.264 and 0.265, respectively, which indicated that the leachate had a low biodegradability in the anaerobic phase of the environment [30]. In 2014, the ratio of $BOD_5/COD_{Mn}$ for leachate (their average was 265 mg/L and 821 mg/L, respectively) ranged from 0.316 to 0.331. In 2017, the ratio of $BOD_5/COD_{Mn}$ for leachate (365 mg/L and 1210 mg/L, respectively) was 0.302. All of the above ratios showed that the continuous low biodegradability environment remained stable. A number of scholars considered that the low concentrations of $COD_{Mn}$ and $BOD_5$ indicated that the MSW landfill was in the intermediate stage of biodegradation because the ratio of $BOD_5/COD_{Mn}$ ranged from 0.1 to 0.5 [31,32], which could be demonstrated by the annual decreasing concentrations of $COD_{Mn}$, $Cl^-$, $SO_4^{2-}$, and TDS from wells 2#, 3# and 4#, as shown in Table 1. The pH values of samples from wells 2#, 3# and 4# rose from 2014 to 2017, indicating that the groundwater environment gradually transformed from weakly acidic to alkaline, which corresponded well with the Cl–Na type designation for the polluted groundwater.

*4.2. Factor Analysis*

According to the results from the factor analysis package of SPSS, four factors with eigenvalues over 1.0 [33] were obtained to reflect the similarities and differences in the original chemical data. Therefore, the difference in groundwater quality could contribute to understanding the combined function of all four factors. The eigenvalues of the four factors were 4.426, 1.880, 1.148 and 1.005, and their variance percentages were 49.181%, 20.890%, 12.752% and 11.162%, with a cumulative variance of 49.181%, 70.071%, 82.824% and 93.985%. Therefore, the groundwater component characteristics could be characterised by the above factors, as seen in Table 2. Four factors were extracted to represent the main factors for groundwater quality, and to identify the different pollution sources.

**Table 2.** Factor analysis resulting from the monitoring data.

| Parameters | Factor Analysis Result | | | |
| --- | --- | --- | --- | --- |
| | 1st Factor | 2nd Factor | 3rd Factor | 4th Factor |
| $NH_3$-N | −0.228 | −0.087 | 0.027 | 0.958 |
| $COD_{Mn}$ | 0.079 | 0.868 | 0.263 | 0.226 |
| $NO_2^-$ | −0.128 | 0.775 | 0.416 | −0.158 |
| Total hardness | 0.949 | −0.199 | 0.199 | 0.055 |
| $Cl^-$ | 0.986 | −0.070 | −0.074 | 0.038 |
| $SO_4^{2-}$ | 0.874 | 0.256 | 0.197 | 0.032 |
| Pb | 0.575 | −0.478 | 0.658 | −0.016 |
| TDS | 0.958 | 0.031 | −0.247 | 0.065 |
| Mn | 0.683 | 0.424 | −0.572 | 0.014 |
| Eigenvalue | 4.426 | 1.880 | 1.148 | 1.005 |
| Variance (%) | 49.181 | 20.890 | 12.752 | 11.162 |
| Cumulative variance (%) | 49.181 | 70.071 | 82.824 | 93.985 |

The first factor, with the principal components of total hardness, $Cl^-$ and TDS (greater than 0.900) and the secondary components of $SO_4^{2-}$, Pb and Mn (ranging from 0.500 to 0.900), contributed approximately 49.181% to the groundwater quality, indicating that most of the groundwater pollution in this study area resulted from the leachate leakage of the MSW landfill, similar to the substantial studies that demonstrated that the main influential groundwater pollutants from landfills included $Cl^-$, $SO_4^{2-}$, total hardness, TDS and trace metals [34–38]. The TDS of this landfill was mainly determined by $Cl^-$ and $SO_4^{2-}$; therefore, $Cl^-$ was very mobile and generally constituted nonreactive tracers, as seen in Figure 6e [18], and $SO_4^{2-}$ underwent far-reaching biochemical transformations [19]. As Figure 6 shows, a high similarity was obtained for the spatial distribution of Pb, total hardness, $Cl^-$, $SO_4^{2-}$ and TDS, as seen in Figure 6c–g. It also shows that the deterioration tendency of the groundwater quality in this study area was a consequence of the lack of a drainage system [35] in Diaoyushan MSW landfill. In the rainy days, most of the polluted surface water affected by the leachate flowed out of the village to the eastern and southern croplands, and the zone of the village showed slight pollution. Mn, as one of the secondary components, was mainly found in the acid-soluble, as seen in Table 2, and reducible soils [39]. Parts of the joint faces were stained a greyish-green colour by Mn oxide. Mn-polluted zones were situated in the west of Dongdasha Village (transitional zone of groundwater) and have a different spatial distribution from Pb. Its transportation and enrichment were restrained by local hydrogeological conditions and the redox environment, and partially affected by the original geologic elements. According to the geological survey in 1991, a slip surface enriched in Mn was covered above the middle Proterozoic Jinping formation, and the lithification test also showed that Mn was found in the bedrock.

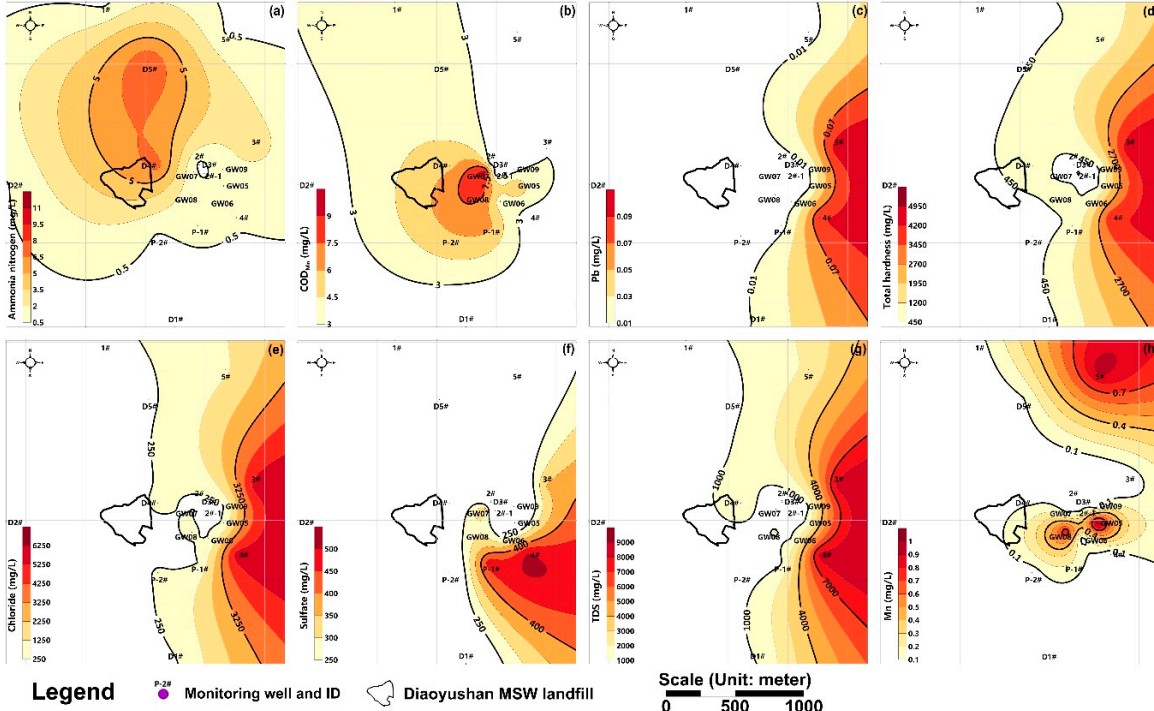

**Figure 6.** Spatial distribution of (**a**) NH$_3$-N, (**b**) COD$_{Mn}$, (**c**) Pb, (**d**) total hardness, (**e**) chloride, (**f**) sulfate, (**g**) total dissolved solids (TDS) and (**h**) Mn.

The second factor, including COD$_{Mn}$ and NO$_2^-$ ions, contributed 20.890% to the groundwater quality. COD$_{Mn}$ is usually characterised as the critical oxidising parameter for determining water quality and showing the degree of organic contamination in water bodies [40]. In this study, the distribution of high concentrations of COD$_{Mn}$ was located at both hillsides of Diaoyushan Hill and Huaiheding Hill, whose pollution sources were random domestic pollution sites near the south acid groundwater environment. According to the comparison of mapping for NO$_3^-$ and NO$_2^-$, it was found that the distribution of the high concentrations of NO$_3^-$ was analogous to the pollution distribution of NO$_2^-$ [41], which was also spread around the west of Diaoyushan Village and its cropland. Only two point samples, GW05 and GW08 (all in the cropland), for NO$_2^-$ exceeded the standard of groundwater quality; however, no samples of NO$_3^-$ were outside the standard. Previous research showed that croplands [42] and the landfill were the two largest pollution sources of reactive nitrogen in the groundwater environment [43]. Therefore, the main factor of NO$_2^-$ was the intensive application of agricultural fertiliser on croplands [44] and domestic sewage. The NO$_2^-$ concentration was low in the majority of the research area, which was similar to the area of lower ratio for COD$_{Mn}$ and NO$_2^-$ ions. Based on the factor analysis data of Table 2, COD$_{Mn}$ showed a positive correlation with NO$_2^-$. It was further illustrated that the area of Dongdasha village was located in the transitional zone of the pollution plume of landfill leachate [45,46].

The third factor, including Pb, contributed 12.752% to the groundwater quality. Previous research considered that trace Pb pollution originates from human activities and has a direct relationship with regional socioeconomic development, rather than the physicochemical characteristics [47]. The area of high Pb concentration was located within or near villages and roads. Pb was found in the nonbioavailable residual form in the shallow surface sediments [40]. High Pb values were not only near Dongdasha Village, but also in the northern villages, as seen in Figure 6c. The concentration of Pb in four soil samples along the landfill boundary showed that the downstream catchment of the landfill was influenced, and they ranged from 19.3 mg/kg to 45.2 mg/kg (downstream area) with an average 32.33 mg/kg. The leachate leakage into the groundwater also increased the content of Pb in the soil. Hence, the absolute eigenvalues of Pb in the first and third factors all exceeded 0.500, indicating

that the concentrations of Pb were not only influenced by the MSW landfill, but also by other human activities and the presence of natural minerals [48]. In the third factor, there is an inversely proportional correlation between Mn and Pb, which illustrated that the geochemical background was not of original influence. Studies of phosphorite mines have shown that trace metals from tailing exposure and the mobilisation of dust particles by wind and rainfall can influence the environmental geochemistry and human health [49,50]. Near the study site is Jinping phosphorite mine, which was one of the first large phosphorite mines in China, and operated from 1919 to 2004 with raw ore production capacity of 1.2 Mt/a. The petrographic test showed that the Pb content of phosphorite ore ranged from $10^{-5}$ to $10^{-4}$. It was concluded that the landfill and human activities, including phosphorite mining and industrial development, were dominant in the third factor.

The fourth factor, including ammonium, contributed 11.162% to the groundwater quality. It was concluded that it might have originated from anthropogenic sources, including domestic sewage and the intensive application of agricultural fertiliser [51], similar to the second factor. The redox potential for samples 1# to 5# in 2014 ranged from 394 mV to 491 mV, and the ammonium ion concentration ranged from 0.10 mg/L to 0.72 mg/L. As Figure 6a shows, the variation of pH, Eh and redox environment made it difficult to confirm the source of ammonium ions in this study area.

Using factor analysis, four hydrogeochemical characteristic factors from samples in the study area were employed to describe the eco-environmental influence on the chemical composition of groundwater and surface water [52] by the Diaoyushan MSW landfill, human activities, surface domestic sewage and agricultural fertiliser. The dominant condition that constrained the groundwater environment of the study area was the Diaoyushan MSW landfill (>49.18%), followed by domestic sewage in villages, application of agricultural fertiliser in croplands, and the occurrence regularity of the original geologic elements in groundwater. It would be helpful to study the area's history and the path of pollution transportation.

### 4.3. Analysis of Contaminant Transportation and Path

As this study showed, the most credible analyses of groundwater contamination zones were achieved by the Nemerow index method in this study area, as seen in Figure 7a,b. Points D5# and 5# with long distances (678 m and 992 m, respectively) were not taken into account because they were situated at the north-western wing of Huaiheding Hill, as seen in Figure 4, the hydraulic relationship was weak, as seen in Figure 7a, and the shallow groundwater pollution of 5# resulted from domestic wastewater of the villages. Polluted zones resulting from Diaoyushan MSW landfill were located in the downgradient groundwater area located in the south-eastern wing of Huaiheding Hill and in direct proximity to the MSW landfill (4#). By the retardation influence of Dongdasha Village situated in the downstream groundwater area, parts of the polluted water resulting from the leachate flowed out of the village into cropland through the N–S drainage canals (bedrock fissure groundwater monitoring wells GW07, GW08 and pond P-1#) and the W–E direction (bedrock fissure groundwater monitoring wells 3#, GW09 and pore groundwater monitoring well GW05).

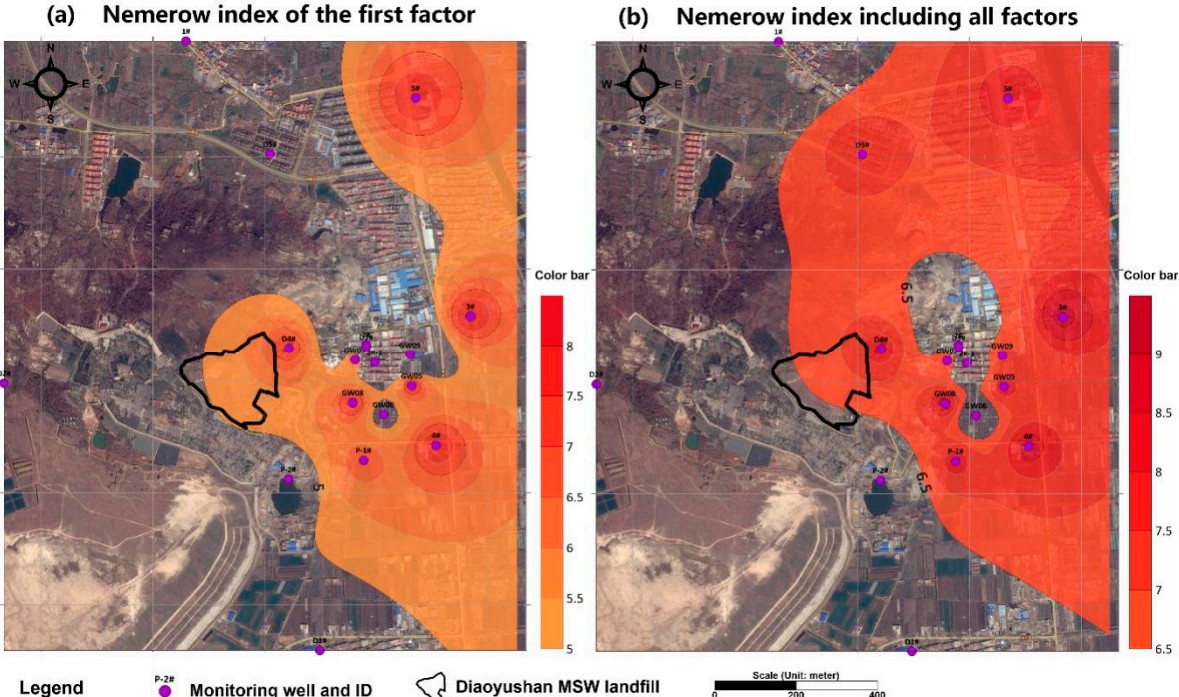

**Figure 7.** Map of the Nemerow index of the study area, including the factors from the factor analysis. (**a**) Nemerow index of the first factor, (**b**) Nemerow index including all factors.

With increasing distance from the landfill, the thickness of the Quaternary sedimentary strata increased. The transition zone where the bedrock fissure groundwater transformed into pore groundwater of the unconsolidated alluvium was regarded as an a priori vulnerable groundwater environment area and suffered other kinds of environmental contamination [18]. The transition zone situated in the north and west of Dongdasha Village, therefore, became the transitional area of contaminant transportation from the landfill. The downstream area of Diaoyushan MSW landfill with relatively poor hydraulic performance became the most polluted zone in this study area, such as the 3# and 4# sample sites, as seen in Figures 6 and 7.

Numerical simulation of solute transport was employed to compare with the result of the Nemerow index and to predict the future path of pollution transport, as seen in Figure 8. Firstly, parameter inversion, adjustment of the simulated model and sensitivity analysis were conducted to obtain the groundwater flow field of the study area. When the system errors between the observed and computed values were acceptable, the area of the landfill and the collecting pond was set at a concentration of 2500 mg/L, then the morphological characteristics of pollution transport were demonstrated as times series, as seen in Figure 8. After 1500 days, only the remaining contaminant grid cells participated in the simulation of solute transport. As the figure shows, the polluted area around the village remained and will continue to expand in the future, but the concentration of excessive chemical contamination will be reduced by natural degradation, as evidenced by the groundwater quality of samples 3# and 4# between March 2014 and July 2017.

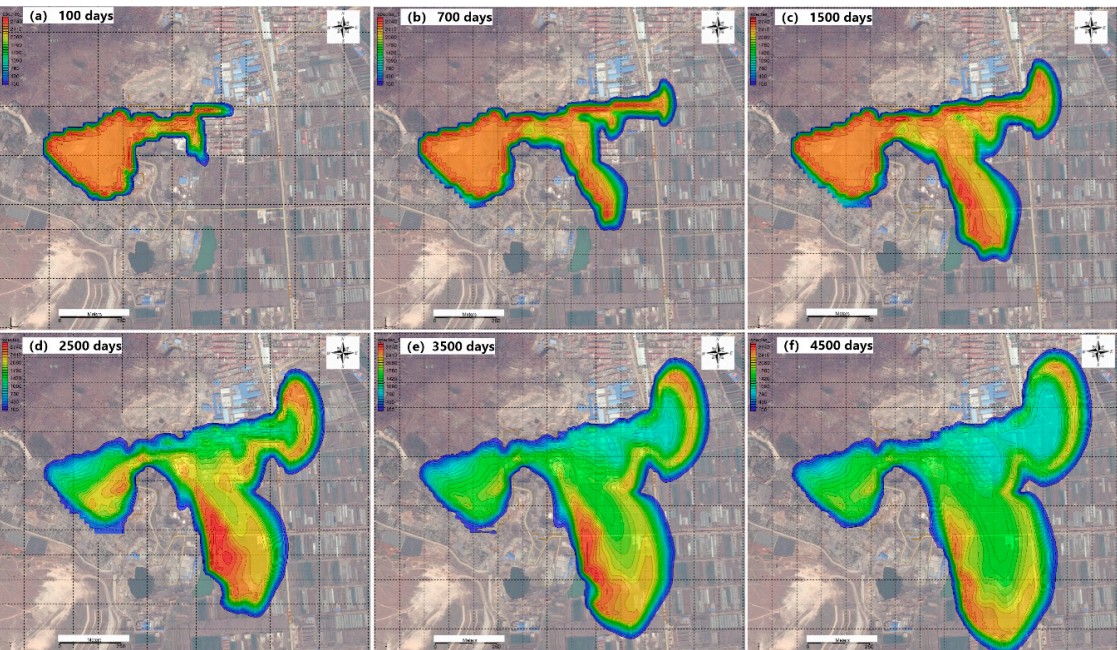

**Figure 8.** Numerical simulation of the leakage of the MSW landfill under the condition of special concentration with 2500 mg/L before 2500 days. The other case assumes no more leakage from the landfill and leachate-collecting pond after 2500 days. Only the remained grid cells of pollution were simulated into solute transport.

From Figures 7 and 8, the leachate leakage during individual years (from 1995 to 2007) of the initial landfill stage was the primary pollution source in the study area due to the absence of a leachate collection system or an intercepting dam. As time progressed, some measurements, including at the leachate collection pond, the pretreatment station, waste dams, the airway tube and a certain number of supporting base measurements, were performed (from 2007 to 2015), and the influence on the groundwater environment from the MSW landfill was greatly diminished, but the main influenced area was still at the south and east of Dongdasha Village. Since the abandonment of the leachate collection pond (2015 to present), some leachate leakage has had an increasing influence on the groundwater environment, but the degree was smaller than in the initial stage. Moreover, simulated solute transport could show the pollution path and predict how the contaminant plumes are transported.

## 5. Conclusions

According to the statistical factor analysis method, the Nemerow index and spatial variations of nine chemical components for analysis of 16 monitoring wells and two ponds along the periphery of the landfill, the leachate leakage from Diaoyushan MSW landfill was the principal pollution source (49.18%) for the groundwater environment. The rainwater and sewage ditches and canals of Dongdasha Village increase transport into the cropland and increase the influence area of pollution. The polluted groundwater area had high concentrations of total hardness, $Cl^-$, $SO_4^{2-}$, TDS and Pb, which all seriously exceed the standard of drinking water quality. The overall groundwater environment was in a phase of natural ecological restoration. To alleviate the contaminant risk and quantify the restoration hierarchy of the local environment in and around the landfill, it is necessary to improve the representative network of monitoring wells, and to trace the evolution and distribution of the polluted groundwater plume by quantitative methods or models, such as construction of new monitoring wells at the south and east of Dongdasha Village and investigation by the Electrical Resistance Tomography method, because the polluted area around the village will continue to expand, but the concentrations of excessive chemical contaminants will be reduced by natural degradation.

**Author Contributions:** Conceptualization software and validation, G.C.; methodology, Y.S., Z.X.; formal analysis and investigation, X.S., Z.C.; writing—original draft preparation, review and editing, G.C., Y.S., Z.X., X.S., Z.C.

**Funding:** This research was funded by National Science Foundation of China, grant number 4F180035 and U1710253.

**Acknowledgments:** This work was supported by the Eco-Environmental Protection Bureau of Lianyungang City, Jiangsu Province. Thanks are also given to the data from the detailed investigation, contaminant resource survey and massive heap stability assessment by The Architechtural Design & Research Institute of Zhejiang University Co, Ltd. The authors would also like to acknowledge the anonymous reviewers for their detailed comments that helped to improve this study.

**Conflicts of Interest:** The authors declare no conflict of interest.

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
