# Peer review of "Assessment of Shallow Groundwater Contamination Resulting from a Municipal Solid Waste Landfill—A Case Study in Lianyungang, China"

_water, doi:10.3390/w11122496_

Round 1

Reviewer 1 Report

General comments

The work contains serious deficiencies and requires significant supplementation and substantial rewording and corrected (preferably by a native speaker). The work presents a local problem and commonly used techniques and research methods. In my opinion the novelty of work is low. The presented by Authors interpretation in many places raises my serious reservations. Therefore, taking into account all above notices, in my opinion, the manuscript in its current form is not suitable for publication. The main reservations are as follows:

1) Insufficient description, and actually the lack of a description of hydrogeological conditions in Holocene/Pleistocene aquifer systems, what in fact prevents the reliable interpretation and formulation of correct conclusions;

2) An important problem is also the amount of data (n = 21) used for interpretation, data are also from various periods (2014 and 2017). The changes in parameters over time in selected sampling points are not unequivocal and do not indicate a well-defined direction of hydrochemical evolution of the investigated groundwater. The authors do not provide the amount of data that was used to conduct the FA. If the all samples from 2014 and 2017 were used for FA, I consider this to be a mistake, as in some wells there are changes in the value of the analyzed parameters over time. As a result, it can significantly distort results, especially with such a little data set.

On the other hand, if only 16 samples were used for the FA (on 2017), this is in my opinion insufficient to carry out a reliable analysis as many as nine variables.

Data are expressed in different units and are also highly variable values - how was the standardization problem solved?

3) There is a lack of discussion or presentation of examples of the use of statistical methods and spatial mapping in hydrogeological studies. An examples of a broader discussion of such methods was provided by Kotowski et al, 2016 a, b or many others.

References:

Kotowski T., Kachnic M., 2016 - The geochemical study of groundwaters from Cenozoic aquifers in the Gwda catchment (Western Pomerania, Poland). Environmental Earth Sciences, 75: 192-207.      doi org/10.1007/s12665-015-4962-x

Kotowski T., Satora S., 2016 - Analysis of the spatial variability of Na and K ions concentrations in the groundwater sourced in the Southern Poland. Environment Protection Engineering, 42(2): 5-18. doi 10.5277/epe160201

Detailed comments regarding specific parts of the text are presented below:

Page 1

Line 12: "… 9 chemical components were collected" - What does actually it mean?

Page 3

Figure 2 is unnecessary in my opinion. Information presented on this figure are reported on other drawings

Page 4

Lines 104-115: Presented hydrogeological parameters ranges (aquifers and landfill) needs citations.

Line 119: Where in the text are the results of these pressure tests?

Page 5

Figure 4. The quality of figure 4 is poor. Some marks are illegible. What means the labels on groundwater contours? The unit of the topography is given in Chinese.

Page 6

Table 1.

How was calculated of distance of sampling points from leachate collecting pond? Have the groundwater circulation system and hydrodynamic conditions been included in the calculations?

What metals were included in the calculation of the Numerow index value?

Page 8

Lines 5-7: "… Each groundwater sample was taken below water level 1.0 m after sufficient pumping commenced to ensure the stability of water quality" - What does it mean? How it was controlled? There is no information at what depth the well screens are located, so actually, what depth were the samples taken from?

Line 11: For furnace waste and slag, the concentrations of other heavy metals should be investigated.

Lines 15-19: Unnecessary description - just refer to one or several papers on this topic.

Page 9

Line 47: Figure 4. Only four samples are contaminated? Table 1 shows that there are far more than four of them.

Page 10

Line 79: I definitely do not agree with the statement that SO4 ions are non-reactive tracers - on the contrary, sulphates undergo far-reaching biochemical transformations (e.g. BSR and others).

Lines 81-82: "… deterioration tendency of the chemical components" - What does it mean? It is unclear.

Line 83: Which water - surface flow or groundwater flow?

Lines 86-90: Could the Authors justify this statement more widely?

Page 11

Figure 6. Has a map of errors and modeling of the semivariogram been made? In the case of random models, the execution of contour maps may be subject to a large error. These maps are significant points used by Authors to final interpretation.

Line 94: "… contributed 20.89% to the groundwater quality." What does this sentence mean? It is totally unclear.

Line 99: "… concentrations of NO3 - this ion was no analyzed by Authors. What does this sentence mean?

Lines 102- 103 "… main factor of NO2- was the intensive application of agricultural fertilizer on croplands and domestic sewage". Could the Authors justify this statement more widely? The NO2 ions concentrations are in majority low in research area (Table 1). How is main  relationship between the COD and  NO2 ions in such low concentrations?

Line 104: Again. See comment to line 94.

Line 104 -107: Is it related to the groundwater? Could the Authors justify this statement more widely?

Lines 109-111: I do not understand the basis for this conclusion based solely on the value of eigenvalues of Pb for factors 1 and 3. Factor 1 relates to the presence of Pb in groundwater as a result of the impact of the landfill. In factor 3 there is inversely proportional corelation Pb with Mn, which the Authors unfortunately do not explain. It is not clear with which process within the factor 3 is correlated of Pb , because the presented interpretation regarding phosphate rock mining is not sufficiently justified.

Line 116: Again. See comment to line 94.

Line  116-118: In factor 4, ammonium ions is not significantly correlated with any other parameter and should not actually be interpreted in my opinion. The  interpretation presented by Authors I consider insufficiently justified, especially taking into account the fact that leachate from the municipal waste landfill is a very important source of NH4 ions. Therefore, without additional research (e.g. isotopic studies, redox conditions etc.), it is not possible to perform a reliable interpretation this phenomena.

Author Response

Dear reviewer,

Thanks for your comments of this manuscript. 

Please see the attachment for point-to-point response.

Reviewer 2 Report

Even the paper is well written and understandable, I suggest major revision as there are some points which should be considered:

Table 1 should be moved to the results and discussion part. why some data collected in 2014 and some in 2017? what about 2015 and 2016? The objectives are unclear and should be revised at the end of the introduction. Don't write "in this paper: better use in this study or research. what about the China standard of groundwater? The limit of each parameter should be added to the paper. What is the author's suggestion or estimation for the coming years about this pollution? It should be added to the conclusion.

Author Response

Dear reviewer,

Thanks for your comments.

Table 1 was moved to the chapter 4.1 of results and discussion.

Due to the discontinuous monitoring and restrained information, parts of groundwater quality data was missed in 2015 and 2016. So as more parameters and factors were taken consideration as possible, we chose the data of 2014 and 2017.

In the end of introduction, “The contribution of landfill to groundwater quality of study area and the mechanism of how to influence the groundwater environment from the initial stage to the present and future will be studied and discussed. ”was added.

ALl the words of “in this paper”,”this paper” or “as this paper showed”were corrected into “study”.

GB/T14848-2017 is the available standard supported for the national groundwater pollution survey and evaluation and used to assess the groundwater quality in China.

The III standard concentration of standard for groundwater quality in GB/T14848-2017 (China) were listed in the last line of Table 1.

The estimation of polluted area in future was simulated and showed in Figure 8, the suggestion was added into the conclusion and marked with red color.

Best Regards.

Chen

Reviewer 3 Report

Figure 1 – The image on the left is designed to provide a regional location; yet, as someone unfamiliar with the area I do not know where the study site is located.  More detail is needed. 

Lines 82-85 (p.3) What is the purpose of these lines?  The information does not appear to provide any relevant information.

Figure 2 – Needs a scale and a legend.  What do the cyan dots represent.  Where is the exact area in reference to Figure 1 as it it does not have the exact coverage of the landfill.

Section 2.2 – The design of the landfill coupled with the geology needs to be better presented.  How is the landfill created in terms of liners?  What is the direct underlying geology?  It appears it is a granite, but confirmation would be good.  Joints in the granite are mentioned to a depth of 15 m.  How deep is the landfill?  Given the size, I suspect it is deeper than 15 m, which would suggest that the joints do not have a significant role – please elaborate.  How thick are the Quaternary sediments and what role do they play?

The 10-7 cm/s K for the granite – is this for the jointed material or the granite in generally?  The value seems high for a granite.  How was it determined? 

Table 1 – Include the data for all of the ions and parameters presented in the work AND for all of the data presented in the figures.  One cannot evaluate the interpretive graphs if the data cannot be examined.

How was CODMn measured/

Results:

I question the interpretation of the data.  If the leachate pond is to be a representative of the landfill, the samples that are considered polluted are suspect.  The leachate pond water has lower values of TDS, Cl, sulfate than the wells that are considered polluted.  If those wells are polluted is it from the landfill?  The values for the polluted wells are much higher than the leachate pond waters.  Some of the wells described as polluted in Figure 5 (5#) are not along groundwater flow paths that would have originated in the landfill. The spatial distribution of the parameters presented in Figure 6 further raise questions about the source.  For a number of the parameters, the concentrations are lower near the landfill than they are as one moves away.  One would suspect that the landfill would be a continuous contributor of solutes; thus, a plume of material would be seen from the landfill.  That is not the case. 

BOD5 is introduced in the results with no mention of how it was measured or what the values observed were.

Author Response

Deart Reviewer,

Reviewer 4 Report

The paper entitled Assessment of shallow groundwater contamination
 resulting from the municipal solid waste landfill : A  case study in Lianyungang, China presents using of a factor analysis and the Nemerow index to evaluate the influence of a municipal landfill on groundwater based on the data from 16 groundwater monitoring wells. Overall the paper sounds good and it is well organized. Unfortunately its novelty is poor. Data series should take into account longer period. Paper describes only one hydrogeological situation. A good solution for this paper is to extend it via modeling research.

Author Response

Dear reviewer,

Thanks for your comment. The numerical simulation of the leakage from MSW landfill was added and showed in this study and related estimation of the remaining polluted area was also conducted. It was treated as the tool to predicate the polluted plume and improve the monitoring network in future in this study.

Best Regards.

Chen

Round 2

Reviewer 1 Report

After re-reviewing the manuscript entitled Assessment of shallow groundwater contamination resulting from landfill of municipal waste: a case study in Lianyungang, China, I have the following comments:

Most of my substantive reservations regarding the interpretation of the results of this work have been taken into account by the Authors, modified and/or more clearly justified, so and in the current form it better illustrate the results of research obtained by the Authors. The manuscript has also been supplemented with new relevant information that allows better understanding of the issues presented. The effort put in by the Authors to improve the work is visible, but work in some elements requires further corrections.
1) In particular, this applies to English. I suggest that this text should be corrected by a native speaker;
2) Only one kind of unit should be used to determine the filtration coefficient. No value conversion to [m/d] is necessary;
3) There are many editing errors, typos and obvious errors - e.g. the expression "national degradation" on line 383. I suggest conducting a very careful editing and improving the manuscript by people who have not read this text before.

Author Response

Thanks for your comments of improving the quality of this manuscript, 1) In particular, this applies to English. I suggest that this text should be corrected by a native speaker; Answer:The manuscript had been polished by a native speaker and the modified context was presented in the manuscript. 2) Only one kind of unit should be used to determine the filtration coefficient. No value conversion to [m/d] is necessary; Answer: The unit of [m/d] was all canceled in this manuscript. 3) There are many editing errors, typos and obvious errors - e.g. the expression "national degradation" on line 383. I suggest conducting a very careful editing and improving the manuscript by people who have not read this text before. Answer: Thanks for your careful reading, the errors in line 313, 316 and 283 were corrected.

Reviewer 2 Report

The revised version looks better but there are some issues. As the authors said the data are missing in 2015 and 2016, how they can connect the obtained data in 2014 to data in 2017? Is it logical as the data are not continuous? How you can justify this missing? Are there any similar published studies that missed 2 years of data between? I think the readers want to see the trends. Also, check the English in whole the manuscript especially the red-highlighted parts by a native speaker. I suggest major revision.

Lines 123-125: The sentence is unclear. Please check.

Lines 126-128: Poor language.  The verb in the first sentence is "is" and in the second sentence is "will be transformed". Check and re-write.

Lines 139-148: Why the authors add these? I think it needs to be summarized.

In the title of Table 1, what is "the III standard concentration of standard for groundwater quality in GB/T14848-2017 (China)"? May you explain?

Some of the data in Table 1 are inconsistent. Why NH 3 -N in Feb 2017 was 14.75? Any explanation? 

Line 281-288 no reference? How the statistics don't have any references?

Line 301-304 and 312-314: poor English.

Line 324: Don't use "we" in the paper.

Check the reference list. The format should be according to the journal.

Author Response

Dear reviewer,

Reviewer 3 Report

The authors have addressed my comments and the additional edits/modification have strengthened the paper.  The addition of the numerical simulation figure (Fig 8) and the corresponding text enhance the results and the interpretation. 

Author Response

Thanks for reviewer’s comments and suggestions for improving the quality of this manuscript. Numerical simulation for pollution plume would be treated as one useful tool for predicate and control plume at the stage of ecological remediation. The academic language and style would be polished and checked.

Reviewer 4 Report

Authors have corrected all comments and suggestions. Paper should be published. 

Author Response

Thanks for reviewer’s comments and suggestions for improving the quality of this manuscript. English language and style was polished and checked carefully.